# Research on the impact mechanism of digital technology on the new-quality productivity of port logistics

Kebiao Yuan [1]*, Zhijian Xu[1], Haiwei Fu[2]

1 School of Economics and Management, Ningbo University of Technology, Ningbo, Zhejiang Province, China, 2 School of Economics and Management, Ningbo University of Technology, Ningbo, Zhejiang Province, China

* ykbjob@163.com

## Abstract

The development of port digital technology is an important way to promote the development of new-quality productivity in port logistics. This study takes Ningbo Port, Shanghai Port, and Tangshan Port as the research objects, and selects the data from 2013–2023 to explore the inner mechanism of the impact of digital technology on the new-quality productivity of port logistics. The results show that digital technology has a significant positive impact on the new quality productivity of port logistics, and the conclusions remain robust after the endogeneity test of the results using the instrumental variables method, eliminating the data of abnormal years, adding fixed effects, and adopting the instrumental variables method. The mechanism test shows that digital technology affects the level of the new quality productivity of port logistics by facilitating the development of new types of digital infrastructures. The digital economy also plays a mediating role between digital technology and the new-quality productivity of port logistics. The findings of the study not only provide key clues to accurately grasp the connotation and characteristics of new productivity in port logistics but also further expand and deepen the breadth and depth of theoretical and empirical studies on the development of digital technology and new-quality productivity in port logistics.

## Introduction

Against the backdrop of the rapid development of the global economy, the concept of "new-quality productivity" has emerged, which not only focuses on the restructuring and optimization of production factors but also emphasizes the comprehensive enhancement of value through innovation and technology drive [1]. As an important hub of global trade and a crucial link in national economic development, port logistics faces a dual challenge in the reconstruction of its productive forces: the traditional operation mode has efficiency bottlenecks and environmental costs, and existing

**Data availability statement:** The data underlying the results presented in the study are available from DOI: 10.13140/RG.2.2.28859.12326.

**Funding:** This research was supported by Research Project of Zhejiang Federation of Humanities and Social Sciences (grant no. 2025N161), Ningbo Philosophy and Social Sciences Planning Project (grant no. G2024-1-06) and Zhejiang Province Philosophy and Social Sciences Planning Project (grant no. 25NDJC136YB).

**Competing interests:** The authors have declared that no competing interests exist.

research lacks a systematic deconstruction of the action mechanism of "digital technology-new-quality productive forces" [2]. There are two major limitations in the existing research theories: firstly, most research focuses on the improvement of a single dimension such as port transportation efficiency, ignoring the systematic changes triggered by digital technology; secondly, it fails to reveal the mediating role of the upgrading of digital infrastructure and the internalization level of technology application during the transformation of new-quality productive forces [3].

In today's era of global industrial change and technological innovation, science and technology innovation have become the core driving force to promote new quality productivity. Among them, the development of digital technology is particularly important, as it has not only reshaped the production and operation mode of many traditional industries but has also become the core driving force to promote new productivity. The port logistics industry needs to take advantage of advanced digital technology to promote the process of port informatization and intelligence to achieve capacity optimization, service enhancement, and ecologically sustainable development. Digital technology is not only an important part of science and technology innovation but also a direct tool to promote the new quality productivity improvement of port logistics. For example, the application of key technologies such as the Internet of Things, big data analysis, and artificial intelligence enables ports to achieve real-time monitoring, intelligent scheduling, and optimized resource allocation, thus significantly improving operational efficiency and service levels.

Although domestic scholars have achieved valuable results in research on the impact of digital technology on the operational efficiency of individual ports, there is a lack of mechanistic research on how digital technology drives the development of new productive forces in port logistics. This paper constructs an evaluation model for the new-quality productive forces of port logistics and digital technology and analyzes the relationship between the two through a benchmark regression with fixed time effects. Moreover, breaking through the perspective of the traditional theory of technical tools, it constructs a theoretical framework of "digital technology penetration-empowerment of new digital infrastructure-internalization of technology application-productivity transformation". Through a dual mediation model, it reveals how new digital infrastructure reconstructs the port information resource network through technologies such as the Internet of Things and 5G networks, and how the process of technology internalization helps port enterprises form the innovative ability to quickly adapt to market changes. The situational analysis of digital technologies such as the Internet of Things, big data, and AI, clarifies the mechanism by which digital technology drives the development of new-quality productive forces from three dimensions: the improvement of the labor skills of port logistics workers, the enhancement of the value of the objects of labor, and the improvement of the efficiency of the means of labor. This provides theoretical support and practical reference for the transformation and upgrading of the port logistics industry. It will also provide theoretical support and practical reference for the transformation and upgrading of the port logistics industry.

## Literature review

### Definition of the concept of new-quality productivity in port logistics

In the eleventh collective study of the Political Bureau of the Central Committee, General Secretary Xi Jinping, when presiding over the study, explained the concept and definition of the new quality of productivity: the new quality of productivity (Nqp) is the advanced quality of productivity in which innovation plays a leading role, breaking away from the traditional mode of economic growth and the path of productivity development, and having the characteristics of high-tech, high-performance, and high-quality in conformity with the new concept of development [4]. The development of new-quality productivity requires deepening reforms in the labor force, labor objects, and labor means: through the talent training of traditional labor force to improve their innovation and working ability; through scientific and technological innovation to discover more new natural objects, more raw materials of technological elements to become labor objects needed for the new-quality productivity; through the development of digitalization to convert more physical material forms or knowledge forms into labor objects [5]. In conclusion, the new quality productivity is a new type of productivity adapted to the development needs of the new era, and its development is of great strategic significance for promoting Chinese-style modernization [6]. We need to vigorously develop new-quality productivity, take scientific and technological innovation as the driving force, and rely on new industries to realize high-quality development and meet the people's needs for a better life.

With the continuous promotion of port-city integration, ports drive the overall economic development of cities. As a key node in the supply chain, the port connects various transportation modes, such as sea and land transportation, to ensure the continuity of logistics. At the same time, port logistics is also essential to China's economic development. Therefore, in the context of modernization and development, developing new-quality productivity of port logistics with science and technology innovation as the core is the main path to achieving high-tech, high-performance, and high-quality development of the port logistics economy [7]. The new-quality productivity of port logistics emphasizes the broad application of new technologies such as digital technology, cloud computing, artificial intelligence, and blockchain. These technologies can not only improve the efficiency of cargo transportation but also enhance transparency and real-time information transmission so that port logistics can realize intelligent management [8]. The embodiment of new-quality productivity in port logistics is through the qualitative improvement and optimization of traditional production factors. Including a high-quality labor force mastering new technologies in port logistics, high-tech and intelligent logistics equipment and systems, as well as the traditional labor objects digital informatization, customization, and personalization to improve the overall efficiency and service quality of port logistics [9]. The new-quality productivity of port logistics promotes in-depth cooperation between ports and upstream and downstream enterprises to form efficient supply chain management. By establishing an information platform to realize the sharing of logistics resources and information, ports can provide more centralized and intelligent services to improve the overall logistics efficiency to meet the changing needs of the social and economic structure [10]. In addition, developing new-quality productivity requires that the new-quality productivity of port logistics should focus on the protection of the ecological environment and realize green logistics while improving logistics efficiency [11].

The formation of new-quality productivity in port logistics needs to realize a leap in the traditional labor force, labor objects, and labor resources, that is, the labor force through the study of professional courses to learn the professional knowledge of port logistics and related skills, through the study of digital technology to master the analysis of logistics data, optimization of routes, management of inventories, and to improve efficiency and so on. From the point of view of labor material, it is necessary to update research and development of high and new technology, intelligent equipment, and systems to give the traditional labor material "new" and "quality" attributes. Port logistics needs to reform the objective of labor. The port was only a place for cargo transshipment. However, the development of new-quality productivity in port logistics requires ports to digitize information, customize services to meet the needs of customers, and develop horizontally to become a value-added service provider.

## Impact of Digital Technology on port logistics

Digital technology (DT) realizes the whole information tracking of goods through the application of the Internet of Things RFID technology and modern information technology, which effectively improves the efficiency of logistics and transportation and makes the whole supply chain transparent and the process simplified [12]. Huang Guiyuan and Li Qiused the DEA model to evaluate the port logistics efficiency. They concluded that there is a gap in the level of port digitalization construction in the Guangxi region, which also leads to low port logistics efficiency compared to the other areas. Hence, the port needs to strengthen the process of digitalization transformation to improve the level of port logistics efficiency [13]. Zhu Chasong et al. assessed the current situation of China's city construction by constructing an evaluation index system and putting forward enhancement strategies. China's ocean center city relies on the development of digital technology, artificial intelligence, the Internet of Things, and other high-tech applications for shipping logistics, which can effectively enhance and develop new-quality productivity in the marine field [14]. The application of digital technology in port logistics has been extended to the field of waterway planning. These technologies have not only changed the traditional way of waterway planning and research but also significantly improved shipping efficiency and transportation levels. In addition, digital technologies have demonstrated their unique advantages in the construction and management of brilliant waterways [15]. Marine weather is highly uncertain. To enhance the resilience of the port supply chain, Xie Feng et al. used digital technology to process a variety of data from meteorology, ports, and geographic information to achieve real-time monitoring of marine weather, thus promoting the specialization, visualization, and intelligent development of marine weather services [16].

The development of new-quality productivity in port logistics mainly relies on the leading role of technological innovation, which pushes it to get rid of the traditional development mode and realize high-tech and high-quality development. Through digital technology, port logistics can innovate in management and operation and improve working efficiency and service levels. The introduction of digital technology enables ports to respond to market changes and customer demand effectively, optimize the allocation of resources, and reduce operating costs. In addition, the development of digital technology also lays the foundation for the development of new-quality productivity in port logistics. It helps the transformation and upgrading of traditional productivity in port logistics.

## Methodology for analyzing the impact of Digital Technology on new-quality productivity in logistics

Port logistics new-quality productivity in the need to focus on the development of logistics service quality, Vishkaei, BM and De Giovanni, P will be VR (virtual reality), artificial intelligence, cloud computing, Internet of Things and other digital technologies from the logistics state, responsiveness, supply chain management and other aspects of the impact of the mechanism on the quality of logistics services [17].Fan. SX divided the influence factors of digital technology on ports into the analysis of logistics technology innovation ability, analysis of logistics considerable data sharing ability, analysis of logistics management upgrading ability, analysis of logistics decision-making transformation ability, and established the EWIF-AHP assessment model to analyze the influence factors of digital technology on logistics enterprises [18]. Burinskiene. A and Daskevic. D used a quantitative analysis hierarchy and statistical data analysis to study the impact of different digital technologies on the logistics industry, and the results obtained showed the importance of digital technologies in enhancing operational efficiency, improving customer experience, and enhancing competitiveness [19]. Zhang Baoyou and others used the DEA (traditional network parcel method), which integrates the standard frontier theory, to formulate an efficiency evaluation model for the logistics industry standards of cities with different levels of development to analyze the influencing factors of the high-quality development of the logistics efficiency of different urban agglomerations [20]. Xifang and other researchers summarized the standard system of smart port: specification guide, supervision and evaluation, IOT perception, essential support, BIM technology, data sharing, production operation, operation management, and comprehensive service from the above multiple perspectives can be evaluated and analyzed for smart port

[21]. The development of new-quality productivity in port logistics requires science and technology innovation at the core. Zhang Ning and Li Shuo used a combination of SEM and fs QCA. They took perception ability, integration ability, and learning ability as mediating variables to study the role mechanism of digital technology on the innovation performance of logistics enterprises [22].

President Xi Jinping proposes to promote the development of new-quality productivity led by technological innovation, which requires the transformation of traditional logistics productivity to new-quality productivity in port logistics [23]. According to existing academic research, the focus is concentrated on analyzing the impact of digital technology on port logistics systems. Therefore, it is imperative to explore the impact of digital technology on the new-quality productivity of port logistics by combining the results of previous research. Specifically focusing on the following aspects: analyzing the impact of informatization and intelligent equipment and systems on the technological innovation ability of port logistics to explore its effect on the enhancement of labor means; evaluating the application of digital technology in enhancing work efficiency and skills training, to study its impact on the transformation and upgrading of the traditional labor force; and examining the effect of the inputs of intelligent logistics equipment in enhancing the precision of loading, unloading, and transportation, to reveal its impact on labor objects. This series of studies will provide more comprehensive and in-depth theoretical support and practical guidance for the development of new-quality productivity in port logistics.

## Theoretical analysis and research hypothesis

### Theoretical explanation of digital technology-enabled port logistics productivity

The transformation of digital technology-enabled traditional productivity to new-quality productivity has brought unprecedented changes to port logistics. The mechanism of digital technology empowering port logistics productivity is shown in Fig 1. From the perspective of the labor force, digital technology has dramatically enhanced the skill level of the port logistics labor force through online learning platforms, virtual training, and other means. Workers can learn at any time and from any place to acquire the latest industry knowledge and skills, which in turn improves their professionalism and operational capabilities [24]. Employees can be trained on core skills such as port machinery operation and intelligent dispatching system application through the simulation platform, which, combined with the learning behavior analysis system, can accurately identify individual skill shortcomings and push personalized learning programs. This "learning by doing" skill internalization mechanism not only accelerates the digestion and absorption of new technologies such as automated loading and unloading equipment and intelligent scheduling systems but also cultivates composite talents with both digital thinking and practical ability, promoting the virtuous cycle of technological progress and human capital upgrading. Secondly, in terms of labor materials, the innovation of digital technology has led to an unprecedented increase in the efficiency of the use of labor materials. Through intelligent and networked systems, ports can achieve efficient integration

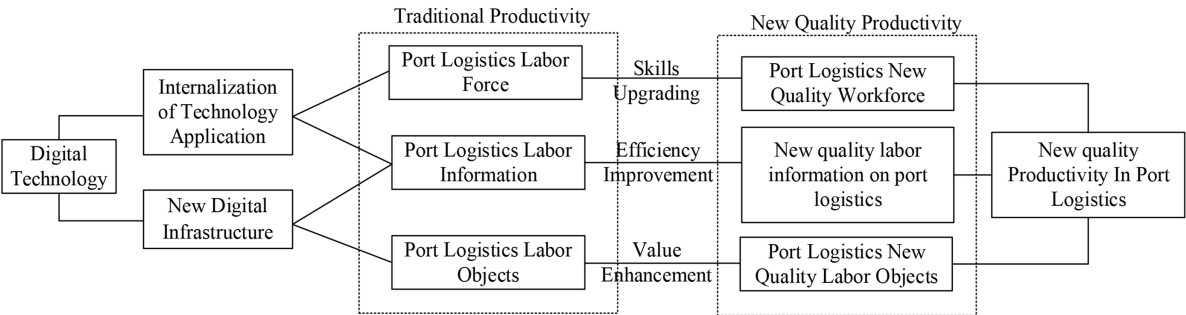

**Fig 1. Mechanisms of digital technology empowering port logistics productivity.**

and utilization of resources to promote the high-quality development of the regional economy [25]. With technologies such as 5G networks, Beidou navigation, and smart sensors, ports can interconnect various types of equipment in conjunction with logistics information systems. This upgrade has given rise to digital infrastructure such as automated container trucks and smart yards, and it has allowed port facilities to develop into an intelligent collaborative network, which can effectively improve the overall operational efficiency of ports. It also lays a good foundation at the physical level for the improvement of the internalization level of technology applications in ports. From the perspective of labor objects, the application of digital technology has changed the work object and service mode of port logistics. The construction of a knowledge network makes port logistics not only rely on traditional physical goods but gradually transform into the flow of information and data. New labor objects, such as digital bills of lading and digital cargo rights certificates, have become the new driving force of port logistics. The introduction of these new labor objects not only improves work efficiency but also enhances service quality so that ports can better meet customer needs. Fig 1 illustrates the mechanism of digital technology empowering port logistics productivity.

In recent years, the wide application of digital technologies has provided a brand new kinetic energy for the productivity leap of port logistics. Brynjolfsson's global port study shows that artificial intelligence and IoT technologies significantly improve the efficiency of port operations through real-time data collection and intelligent decision-making, revealing the direct driving effect of technology on productivity [26]. Further, Zhao H et al. demonstrated, based on a case study of a Chinese coastal port, that blockchain technology validates the optimization mechanism of digital technology on efficiency at the micro level by enhancing supply chain transparency and collaboration [27]. At the infrastructure level, O. F. Andriani et al. found that the extensive coverage of 5G networks can ensure the quality of network connectivity within the entire port area and provide stable communication for loading and unloading [28]. In addition, simulation experiments by P. Aivaliotis showed that digital twin technology can realize fault diagnosis and predictive maintenance of equipment employing physical simulation and artificial intelligence, thus reducing the equipment failure rate [29]. Finally, Chenyu Zhao pointed out that the enterprise technology revolution can effectively improve the total factor productivity of labor-intensive enterprises. Based on the above analysis, digital technology affects the new-quality productivity of ports by penetrating the production process, optimizing resource allocation, and enhancing system stability [30]. Therefore, the hypothesis is proposed:

H1: Digital technology has a significant impact on the development of new-quality productivity in port logistics.

## Mediating role of new digital infrastructure

The efficient transformation of digital technology relies on the physical support and synergistic adaptation of new infrastructure. Yan Xinping et al. found that digital technology realizes the dynamic monitoring of port infrastructure, port environment, and operation status through the deployment of intelligent sensing devices (e.g., IoT sensors and remote monitoring systems), which enhances the automated monitoring capability of ports [31]. This intelligent upgrade not only improves the operational efficiency of the port but also provides fundamental support for the development of new-quality productivity. At the arithmetic support level, Guo Jianke and Yu Fushengqi found that perfect port infrastructure is a core indicator of a city's port logistics and transportation capability, which can provide hardware support for expanding overseas markets, attract logistics enterprises to cluster, enhance the attractiveness of the city and its network status, and strengthen logistics ties with other coastal cities [32]. At the level of synergy, Wan Yu et al. used the SBM-DEA model to study the operational efficiency of ports along the Yangtze River Economic Belt, and the results showed that the level of infrastructure in ports has a significant positive impact on the overall operational efficiency of ports [33]. The new digital infrastructure promotes the deep integration of port, industry, and city by optimizing the information flow between port, industry, and city. Xifang et al. pointed out that digital technology can develop new types of services, such as port logistics and finance, industrial supply chain management, etc., and provide new growth points for the development of ports, industries, and cities, and this integration not only enhances the comprehensive service capacity of ports but also promotes the

high-quality development of the regional economy [21]. Si Zengchuo takes Lianyungang Port and Rizhao Port as empirical evidence, analyzes the connotation between the digital construction of port infrastructure and port economic development, and discovers the new mechanism between the new digital infrastructure of ports and port logistics economy [34]. Lu Jiang et al. proposed that promoting the construction of new digital infrastructure is an important prerequisite for the research and development of digital technology in the promotion, and through benchmark regression analysis, concluded that the new digital infrastructure has a significant positive impact on the development of new quality productivity in the city [35]. Based on the above analysis, the hypothesis is proposed:

H2: The new digital infrastructure has a mediating role in the relationship between digital technology and the new-quality productivity of port logistics.

### The mediating role of the internalization of technology application

The internalization of technology applications can reflect the integration of digital technology into the company's operational system, as well as the employees' mastery of the operation and cognition of the relevant technology in their work [36]. With the continuous development of digital technology, the labor objects of port logistics will undergo iterative upgrading, and their operation principles and operation modes will change accordingly. The internalization of technology application requires that the skill level of the labor force matches the improvement of the efficiency of the labor material to ensure that the labor force can master the new type of labor object and improve the operational efficiency of port logistics. The role of internalization of technology application is reflected in the optimization of port logistics business processes, ports to improve the efficiency and accuracy of business processes, and digital technology inside, which can enable the process to achieve automation and intelligence. The enhancement of port innovation relies on the internalization of technology applications, the internalization of technology applications promotes the port research and development of new products and services but also requires employees to continue to learn and master new technical knowledge and skills, these changes caused by the internalization of technology application can effectively help the transformation of the port logistics new quality productivity, the staff skilled in the application of technology, but also to improve their work efficiency and quality, given the previous analysis, the hypothesis is proposed:

H3: The internalization of technology applications mediates the relationship between digital technology and new-quality productivity in port logistics.

## Research design

### Sample selection and data sources

This paper selects Ningbo Zhoushan Port, Tangshan Port, and Shanghai Port as samples from 2013–2023, whose throughput accounts for more than 30% of the country's total, so their operation scale and market position can effectively reflect the overall development trend of China's ports. The digital transformation process of these three ports is at the forefront of the industry, and it is representative of both the scale and exemplary of the exemplary technology, so these three ports are chosen as samples. The sample data in this paper are processed as follows: the samples with serious missing financial data are excluded, and a small number of samples are linearly filled. The digital technology and other related data in this paper come from the city yearbook and the EPS database. The financial data of port enterprises comes from the annual reports of ports, the China Port Statistical Yearbook, and the CSMAR database.

### Description of variables

**Explanatory variables.** Digital technology development level. This paper draws on the methods of Liu Jun et al. [37] and Yang Junge et al. [38] and constructs the evaluation index system of the level of development of digital technology in the city from the three dimensions of digital informatization, interconnection of networks and digitization of facilities (Table 1).

**Table 1. Construction of the evaluation index system for the level of digital technology development.**

| Level 1 indicators | Secondary indicators | Selection Criteria |
|---|---|---|
| Digital informatization | Number of domain names (million) | Measuring the occupancy of network identification resources |
| Network interconnection | Size of international Internet access subscribers (million) | Assesses the underlying capacity to participate in the global digital economy |
| | Cell phone penetration rate (%) | Characterize the breadth of mobile network coverage |
| | Internet broadband access subscriber size (million) | Measuring the basic service capacity of fixed networks |
| Digitization of facilities | Digitization results | Quantify the effectiveness of the physical application of digital technology |
| | Digital Transformation Index | Assessing the level of industrial digital synergy |

The three-dimensional level 1 indicator covers the whole chain of "resources-connections-applications", with digital informatization indicating the reserve of digital assets (supply side), network interconnectivity measuring the connectivity (circulation side), and digitization of facilities reflecting the effectiveness of applications (demand side). The number of domain names used in the secondary indicators reflects the activity of digital identity of regional economic subjects, which is the core characterization of digital assets precipitation; the scale of international Internet access users determines the efficiency of cross-border data circulation, which is a key indicator for building an open digital hub; the cell phone penetration rate can reflect the fairness of digital access in the society in one aspect, and provide the underlying technical support for the Internet of Things, mobile payment and other scenarios; the density of fiber optic users has a direct impact on cloud computing and application of port logistics. The density of fiber optic users directly affects the quality of development of new types of business such as cloud computing, telecommuting, and online learning in port logistics; the digitalization results reflect the depth of integration of digital technology and physical facilities; the digital transformation index is constructed based on composite indicators such as the enterprise cloud rate and the access rate of the industrial Internet platform, reflecting the maturity of the digital ecosystem of the city. Finally, the weight of each index is determined by the entropy value method, and finally, the index of the digital technology development level of this city is obtained.

**Explained variables.** Port logistics new-quality productivity level. This paper draws on the methods of Zhang Gaohan et al. [39], Yuan Weihai, and Zhou Jianpeng [40]. Unlike the traditional research perspective that solely focuses on operational efficiency, this paper unfolds from the dimensions of three core production factors: labor, means of labor, and objects of labor, and selects eight specific indexes based on the availability of data (Table 2). The level 1 indicator adopts intangible labor data to reflect the intensity of the port's investment in intangible assets such as patents, software, brands, etc., which is the core characterization of innovation-driven productivity; the fixed labor data directly reflects the comprehensive processing capacity of the port logistics system, which also affects the process of automation and intelligent transformation; high-quality talents are the human capital basis for promoting digital transformation and technological innovation; the financial risk indicator reflects the enterprise's investment The financial risk indicator reflects the enterprise's investment capacity, which affects the introduction of new technologies and facility upgrading; the future development indicator reflects the agility of the port logistics system and its ability to optimize the operation of the existing assets, which is highly compatible with the cost reduction and efficiency enhancement goals required by the new quality productivity. To avoid the bias of weight allocation due to subjective judgment, the entropy value method is used to determine the weight of each index and then calculate the index of the new-quality productivity level of the sample ports.

**Control variables.** Referring to existing studies, the control variables selected in this paper include port size (Size) (the natural logarithmic value of total assets), the gearing ratio (ALR) (the ratio of total liabilities to total assets), growth rate of total operating income (GIR) (the ratio of the difference between the current year's and the previous year's

Table 2. Construction of an evaluation index system for new-quality productivity of port logistics.

| Consideration | Level 1 indicators | Secondary indicators | Selection Criteria |
|---|---|---|---|
| Labor resources | Intangible labor resources | Intangible assets as a percentage (%) | Measurement of intellectual capital and technology reserves |
| | Fixed labor resources | Cargo throughput (tons) | Evaluate the scale of port infrastructure operations and resource mobilization capabilities. |
| | | Fixed assets as a percentage (%) | Quantify the intensity of port infrastructure investment |
| Labor force | Quality of employees | Proportion of highly qualified employees (%) | Measure the level of human resources knowledge structure |
| | | Proportion of technical staff (%) | Measure the level of professional and technical talent reserves |
| Target audience | Financial risk | Equity multiplier | Evaluate the stability of capital structure and financial leverage risk |
| | Future development | Inventory turnover (%) | Measure supply chain response efficiency and resource utilization |
| | | Fixed asset turnover (%) | Quantify the effectiveness of facility and equipment utilization |

operating income to the previous year's operating income), return on assets (RA), and the city's level of economic development (GDP) (the city's GDP) [3,5,17,21].

## Modeling

This study adopts a baseline regression analysis model, which can effectively control individual heterogeneity, such as port location advantage and policy additions to avoid estimation errors caused by omitted variables. There may be a bidirectional causal relationship between digital technology and new quality productivity of port logistics, and the use of a fixed-effects model can eliminate individual effects that do not change over time and alleviate some endogenous disturbances through de-meaned treatment. To investigate whether digital technology has a significant effect on the level of new-quality productivity in port logistics and to verify H1, this paper constructs the following benchmark regression model (1):

$$Nqp_{it} = \alpha_0 + \alpha_1 DT_{it} + \sum Control_{it} + \sum Year + \varepsilon_{it} \tag{1}$$

To verify H2, the following regression models (2)(3) are constructed in this paper:

$$NDI_{it} = \alpha_0 + \alpha_1 DT_{it} + \sum Control_{it} + \sum Year + \varepsilon_{it} \tag{2}$$

$$Nqp_{it} = \alpha_0 + \alpha_1 NDI_{it} + \alpha_2 DT_{it} + \sum Control_{it} + \sum Year + \varepsilon_{it} \tag{3}$$

To verify H3, the following regression models (4)(5) are constructed in this paper:

$$ITA_{it} = \alpha_0 + \alpha_1 DT_{it} + \sum Control_{it} + \sum Year + \varepsilon_{it} \tag{4}$$

$$Nqp_{it} = \alpha_0 + \alpha_1 ITA_{it} + \alpha_2 DT_{it} + \sum Control_{it} + \sum Year + \varepsilon_{it} \tag{5}$$

Where: the subscripts $i, t$ represent the port and year, respectively, $Nqp_{it}$ is a proxy variable for the new-quality productivity of port logistics, $DT_{it}$ is the level of digital technology development of the port $i$ in $t$, $NDI_{it}$ is the level of the new digital infrastructure of the port $i$ in $t$, $ITA_{it}$ is the level of the internalization of technology application where the port $i$ is located in

*t*, *Control* is a series of control variables at the port as well as city level, *Year* represents year fixed effects, $\varepsilon$ represents a random perturbation term, $\alpha_0$ represents a constant term, and $\alpha_1\alpha_2$ represent the estimated coefficients.

## Analysis of empirical results

**Descriptive statistics.** Table 3 reports the descriptive statistics of the regression sample used in this paper, with a mean of 1.501 and a standard deviation of 0.320 for new-quality productivity (Nqp) in port logistics and a mean of 1.416 and a standard deviation of 0.592 for digital technology (DT), suggesting that the sample in this paper has fluctuations within a reasonable range for regression analysis.

## Benchmark regression analysis

Table 4 reports the results of the baseline regression of the impact of digital technology on the new-quality productivity of port logistics. Column (1) reports the results of the benchmark regression with only year-fixed effects and individual fixed effects, and Table 4 shows that the coefficient of the impact of digital technology is 0.527, which indicates that the

**Table 3. Sample and its descriptive statistics.**

| Variant | Sample size | Minimum value | Maximum values | Average value | Standard deviation | Upper quartile |
|---|---|---|---|---|---|---|
| Npq | 33 | 0.939 | 2.176 | 1.501 | 0.320 | 1.445 |
| DT | 33 | 0.213 | 2.294 | 1.416 | 0.592 | 1.395 |
| ALR | 33 | −2.484 | 1.747 | −0.000 | 1.000 | 0.223 |
| GIR | 33 | −2.224 | 2.336 | 0.000 | 1.000 | 0.018 |
| GDP | 33 | −0.943 | 2.162 | 0.000 | 1.000 | −0.480 |
| Size | 33 | −0.943 | 2.162 | 0.000 | 1.000 | −0.480 |
| RA | 33 | −1.456 | 2.258 | 0.000 | 1.000 | −0.079 |

**Table 4. Benchmark regression of digital technology affecting new-quality productivity in port logistics.**

| Varianble | -1 | -2 | -3 |
|---|---|---|---|
| | Npq | Npq | Npq |
| DT | 0.527*** | 0.629*** | 0.747*** |
| | -7.418 | -4.787 | -7.841 |
| ALR | | - | 0.046 |
| | | (-−0.274) | -1.161 |
| GIR | | 0.001 | 0.016 |
| | | -0.04 | -0.414 |
| Size | | - | - |
| | | (-−2.249) | (-−1.828) |
| RA | | - | - |
| | | (-−0.267) | (-−1.365) |
| GDP | | | 0.317 |
| | | | -1.34 |
| _cons | 0.755*** | 0.612*** | 0.444*** |
| | -7.115 | -3.261 | -3.226 |
| Year fe | Yes | Yes | Yes |
| N | 33 | 33 | 33 |
| $R^2$ | 0.131 | 0.215 | 0.212 |

coefficient of digital technology is significant and positive at the 1% level, validating H1. Column (2) adds the firm-level control variables, and the results of the benchmark regression show that the coefficient of digital technology is 0.629 and is significant and positive at the 1% level. The coefficient of digital technology is 1.114 and significant and positive at the 1% level, again validating H1. Column (3) adds the city-level control variable, and the coefficient of digital technology is 0.747, still significant at the 1% level. This shows a significant positive effect of digital technology on the new-quality productivity of port logistics, which implies that as the application of digital technology in the field of port logistics deepens, the new-quality productivity of port logistics will increase significantly.

### Endogeneity test

**Instrumental variable method test.** Referring to Yang Peng's approach (2024), this study adopts the text analysis method by searching, matching, and counting the word frequency of the keywords related to "digital technology" in the annual reports of listed companies according to the breadth of digital technology application, and expanding the sample in the year of study [41].In this study, a two-stage least squares regression analysis was conducted to analyze the endogeneity of digital technology affecting new-quality productivity in port logistics by applying the breadth of technology adoption as an instrumental variable. In Table 5, it is shown that the instrumental variable is significantly and positively related to digital technology in the first stage. In the second stage regression, the coefficient of the effect of digital technology on new-quality productivity is 0.7860 and significant at 5% significance level, which indicates that the conclusion that digital technology significantly enhances the new-quality productivity of port logistics is valid. In addition, the instrumental variable passes the test of the LM statistic and F statistic, which indicates that it is not a weak

**Table 5. Endogeneity test of digital technology affecting new-quality productivity in port logistics.**

| | (1) | (2) |
|---|---|---|
| Varianble | First phase | Second phase |
| | DT | Npq |
| IV | 0.1378 | |
| | (1.49) | |
| ALR | −0.0454 | 0.0467 |
| | (−0.87) | (1.42) |
| GIR | −0.0063 | 0.0224 |
| | (−0.09) | (1.47) |
| GDP | −0.7883*** | 0.3462 |
| | (−3.91) | (1.02) |
| Size | 1.1172*** | −0.5285 |
| | (5.84) | (−1.17) |
| RA | −0.2050* | −0.0403 |
| | (−2.06) | (−0.67) |
| DT | | 0.7860** |
| | | (2.29) |
| _cons | 1.0296*** | 0.7916** |
| | (14.07) | (2.34) |
| Year fe | YES | YES |
| Kleibergen-Paap rk LM | | 5.51*** |
| Cragg-Donald Wald F | | 19.71 |
| N | 33 | 33 |
| R² | | 0.903 |

instrumental variable, nor does it suffer from over-identification. Although other control variables, such as gearing ratio, gross operating revenue growth rate, level of urban economic development, and port size, do not have a significant effect on new-quality productivity, return on assets has a significant positive effect on new-quality productivity. The time-fixed effects included in the model help to control for the effect of the time trend and improve the accuracy of estimation. In summary, the results of this study rule out the endogeneity problem.

**Test of bidirectional causality.**  To avoid the endogeneity problem caused by the reverse impact of the explained variable on the explanatory variable, this paper employs the test of bidirectional causality to determine whether there is an endogeneity issue in the benchmark regression analysis. The relevant results were reported in Table 6. Meanwhile, if the impact of the explanatory variable on the explained variable is significant in the original regression, but the impact in the bidirectional causality test is insignificant or the coefficient is extremely small, it indicates that the causal direction of "digital technology affecting the new-quality productivity of port logistics" is more likely to hold, thus verifying H1.

## Robustness tests

**Tests to remove data from outlier years.**  Due to the large shocks to economic development in the epidemic years, the model is retested in this paper after removing the 2010–2021 data sample. Table 7 reports the robustness regression results after removing the data from the outlier years, which shows that the regression coefficient for digital technology is significantly positive at the 1% level.

**Tests incorporating additional fixed effects.**  Table 8 reports the results of the regression test for the inclusion of more over-fixed effects. Where column (1) reports the results of including both year and individual fixed effects, the coefficient on digital technology is seen to be 0.300 and significant at the 1% level. Column (2) adds the firm-level control variable, and the results show that the coefficient on digital technology is 0.192 and significant at the 1% level. Column

**Table 6. Test of bidirectional causality between digital technology and new-quality productivity of port logistics.**

| Varianble | (1) |
|---|---|
| | DT |
| Npq | 0.119 |
| | (0.3740) |
| ALR | 0.023 |
| | (0.0615) |
| GIR | 0.045 |
| | (0.7009) |
| GDP | −0.015 |
| | (−0.2497) |
| Size | 0.020 |
| | (0.5403) |
| RA | −0.263 |
| | (−0.8796) |
| _cons | 0.846 |
| | (0.7463) |
| year fe | Yes |
| Id fe | Yes |
| N | 33 |
| R² | 0.953 |

**Table 7. Robustness test results excluding outlier years.**

| Varianble | (1) |
|---|---|
| | Npq |
| DT | 0.796*** |
| | (4.503) |
| ALR | 0.019 |
| | (0.246) |
| GIR | −0.006 |
| | (−0.092) |
| GDP | 0.337 |
| | (1.251) |
| Size | −0.520 |
| | (−1.711) |
| RA | −0.021 |
| | (−0.255) |
| _cons | 0.479* |
| | (1.988) |
| year fe | Yes |
| N | 27 |
| R² | 0.514 |

**Table 8. Robustness test results with the inclusion of more fixed effects.**

| Varianble | (1) | (2) | (3) |
|---|---|---|---|
| | Npq | Npq | Npq |
| DT | 0.300*** | 0.192*** | 0.178** |
| | (3.283) | (8.251) | (2.667) |
| ALR | | 0.094*** | 0.089*** |
| | | (4.592) | (2.835) |
| GIR | | 0.091*** | 0.088*** |
| | | (6.444) | (7.241) |
| GDP | | −0.058 | −0.002 |
| | | (−1.435) | (−0.011) |
| Size | | −0.152*** | −0.144** |
| | | (−4.515) | (−2.251) |
| RA | | | −0.054 |
| | | | (−0.256) |
| _cons | 1.077*** | 1.230*** | 1.249*** |
| | (16.484) | (17.148) | (10.586) |
| year fe | YES | YES | YES |
| id fe | YES | YES | YES |
| N | 33 | 33 | 33 |
| R² | 0.308 | 0.602 | 0.603 |

(3) adds the city-level control variable, and Table 8 shows that the coefficient of digital technology is 0.178, still significant at the 5% level, which shows that digital technology can improve the level of new-quality productivity in port logistics and also shows that the core conclusion of this paper is correct.

**Test for lagging explanatory variable by one period.** The results of the robustness test using digital technology lag one period as a new explanatory variable are reported in Table 9. The results show that the coefficient of digital technology lagged one period is 0.762 significantly positive at the 1% level, and the above findings are consistent with the previous section, and the regression results pass the robustness test.

## Mechanism testing

Digital technology enhances the new-quality productivity level of port logistics through the improvement of the level of new digital infrastructure. In this paper, we draw on the methodology of Xiao-Jing Banknote et al. [42], using Python software to process the new digital infrastructure-related terms in the government work report and calculate the new digital infrastructure-related percentage to reflect the level of new digital infrastructure in each region. Columns (1) and (2) of Table 10 show the results of the test of the level of new digital infrastructure as a mediating variable, and the results show that digital technology has a significant positive effect on new digital infrastructure, while new digital infrastructure also has a positive effect on new-quality productivity at the 5% significant level. This indicates that digital technology has an indirect positive impact on the new-quality productivity of port logistics through the mediating variable of new digital infrastructure.

The role of internalization of technology application as a mediating variable is also validated from the results in columns 3 and 4 of Table 10. Digital technology has a significant positive effect on technology application internalization (coefficient of 0.055, t-value of 2.672, and significance level of 5%), while digital technology also has a significant positive effect on new quality productivity through technology application internalization (coefficient of 0.643, t-value of 3.602, and significance level of 1%). This further suggests that digital technology has an indirect positive impact on the new quality productivity of port logistics not only through the new digital infrastructure but also through the mediating variable of internalization of technology applications [43].

The impact of digital technology on the new-quality productivity of port logistics is significant, and this impact is further validated through the mediating variables of new digital infrastructure and internalization of technology applications. The structural equation model diagram for mechanism test is shown in Fig 2. This indicates that the development and application of digital technology can not only directly enhance the new-quality productivity of port logistics but also indirectly

Table 9. Robustness test results for replacement explanatory variables.

| Variable | Npq |
|---|---|
| _cons | 0.819*** |
| | (5.8266) |
| LV | 0.762*** |
| | (7.7369) |
| ALR | 0.053 |
| | (1.0072) |
| GIR | 0.076** |
| | (2.8458) |
| GDP | 0.422** |
| | (2.6656) |
| Size | −0.589*** |
| | (−3.1659) |
| RA | −0.085* |
| | (−1.8318) |
| year fe | YES |
| N | 33 |
| R² | 0.905 |

**Table 10. Results of the mechanism of action tests.**

| Varianble | (1) | (2) | (3) | (4) |
|---|---|---|---|---|
| | NDI | Npq | ITA | Npq |
| DT | 0.236*** | 0.539** | 0.055** | 0.643*** |
| | (3.724) | (2.767) | (2.672) | (3.602) |
| NDI | | 0.880 | | |
| | | (1.564) | | |
| ITA | | | | 1.874 |
| | | | | (1.049) |
| _cons | −0.103 | 0.534** | 0.154*** | 0.155 |
| | (−1.138) | (2.528) | (5.204) | (0.447) |
| controls | YES | YES | YES | YES |
| year fe | YES | YES | YES | YES |
| N | 33 | 33 | 33 | 33 |
| $R^2$ | 0.421 | 0.285 | 0.822 | −0.053 |

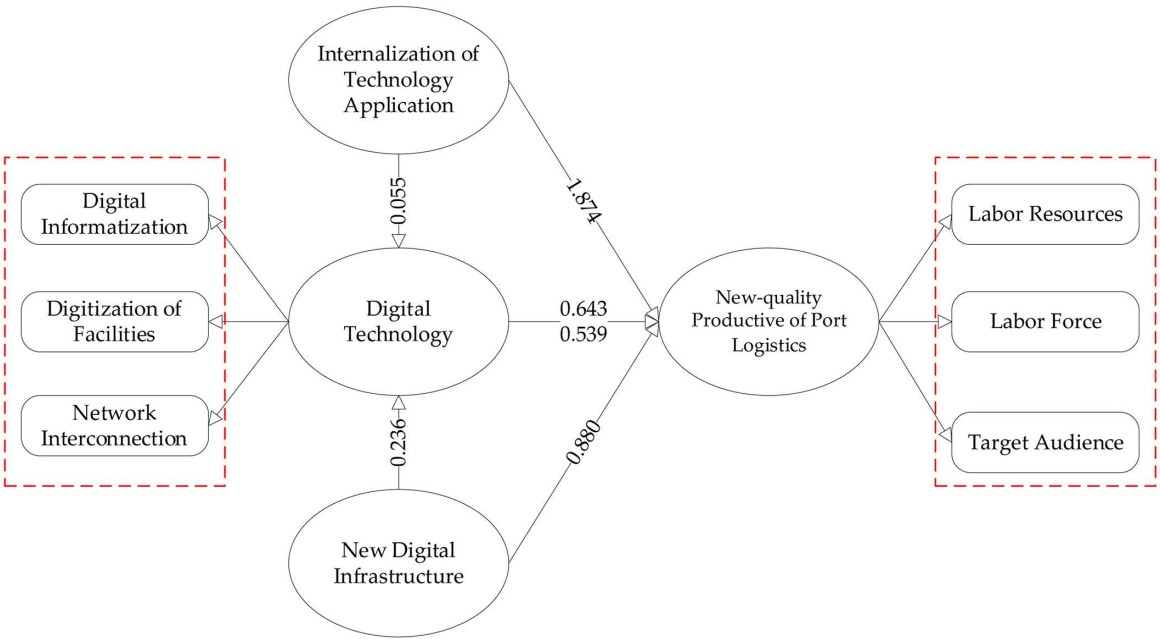

**Fig 2. Structural equation model (SEM) diagram of the mechanism test.**

enhance the new-quality productivity of port logistics by promoting the development of new digital infrastructure and internalization of technology applications. Hypotheses H3 and H4 were validated.

## Conclusions and recommendations

With the development of the global economy, digital technology has gradually become a new engine for the development of new-quality productivity. The continuous development of digital technologies such as the Internet of Things, big data, and cloud computing promotes the transformation of enterprise labor, labor materials, and labor objects to new-quality

productivity. This paper takes Ningbo Port, Shanghai Port and Tangshan Port as research samples and analyzes the data from 2013–2023 and finds that: (1) digital technology has a significant positive impact on the new-quality productivity of port logistics and the conclusions remain robust after eliminating the data of abnormal years, adding fixed effects, and adopting the instrumental variables method; (2) The new digital infrastructure plays a mediating role in the process of digital technology influencing the new-quality productivity of port logistics. Specifically, digital technology indirectly enhances the new-quality productivity of port logistics by facilitating the development of new digital infrastructure; (3) the digital economy also plays a mediating role between digital technology and the new-quality productivity of port logistics. The development of the digital economy can not only directly enhance the new productivity of port logistics but also indirectly enhance the new productivity of port logistics by promoting the development of the digital economy.

Based on the above conclusions, this paper puts forward the following policy recommendations:

From the perspective of the dimension of labor data, at the level of integrated planning of digital infrastructure, ports need to take the lead in building an area-wide IoT sensing system, prioritizing the deployment of 5G+Beidou positioning system, and realizing the centimeter-level collaborative operation of AGVs, unmanned drawbridges, smart yards, and other equipment. For example, Meishan Terminal of Ningbo Zhoushan Port is fully connected through end-to-end IoT to realize all elements can be sensed and digitized, and the port brain will take data as a strategic resource throughout the whole process of the whole operation chain; to promote the informatization and intelligent process of port logistics, such as real-time monitoring, intelligent scheduling, and optimal allocation of resources, which can effectively improve the operation efficiency and service level; and the enterprises are actively constructing and improving the port logistics Enterprises actively build and improve port logistics information platforms, such as intelligent gate systems and intelligent scheduling systems, to achieve the common sharing of logistics resources and information, and enhance the overall efficiency of logistics.

In terms of workforce dimension, to address the digital skills gap, ports should take the lead in forming an industry-education integration community, it is necessary to establish a knowledge updating system in line with the development of digital technology, and regularly push special micro-courses through enterprise digital learning platforms, so that the application of new tools such as smart port operating system can be effectively internalized, which will enhance the adaptability of the staff to the new technology and operation level, thus improving the overall work efficiency; policymakers need to improve the policy of the port logistics system, to enhance the overall logistics efficiency. Work efficiency; policymakers need to improve the digital skills certification system, incorporate new job types such as remote control of gantry cranes, intelligent collector truck manipulation, remote control of bridge cranes, and operation of intelligent cargo handling systems into the national vocational qualification catalog, and issue skills subsidies to employees who have completed their digital learning, to stimulate the internal driving force of workforce transformation.

In the dimension of the labor object, policymakers should promote the comprehensive electronic port operation documents to help reconstruct the port business process; for the innovation of logistics services, the government needs to formulate policies to support the port logistics enterprises to carry out product and service innovation with the help of digital technology, such as the development of new logistics coping solutions, and the provision of customized logistics services.

## Author contributions

**Formal analysis:** Kebiao Yuan, Zhijian Xu, Haiwei Fu.

**Funding acquisition:** Kebiao Yuan, Haiwei Fu.

**Investigation:** Kebiao Yuan, Zhijian Xu, Haiwei Fu.

**Methodology:** Kebiao Yuan.

**Project administration:** Kebiao Yuan.

**Resources:** Zhijian Xu.

**Supervision:** Kebiao Yuan.

**Writing – original draft:** Kebiao Yuan.

**Writing – review & editing:** Kebiao Yuan.

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
