## [Decision Letter · Decision Letter 0]

28 Mar 2025

PONE-D-25-08375Research on the Impact Mechanism of Digital Technology on the New-Quality Productivity of Port LogisticsPLOS ONE

Dear Dr. FU,

Thank you for submitting your manuscript to PLOS ONE. After careful consideration, we feel that it has merit but does not fully meet PLOS ONE’s publication criteria as it currently stands. Therefore, we invite you to submit a revised version of the manuscript that addresses the points raised during the review process.

We look forward to receiving your revised manuscript.

Kind regards,

Xu Xin

Academic Editor

PLOS ONE

Journal Requirements:

This research was supported by Research Project of Zhejiang Federation of Humanities and Social Sciences (grant no. 2025N161), Ningbo Philosophy and Social Sciences Planning Project (grant no. G2024-1-06) and Zhejiang Province Philosophy and Social Sciences Planning Project (grant no. 25NDJC136YB).

This research was supported by Research Project of Zhejiang Federation of Humanities and Social Sciences (grant no. 2025N161), Ningbo Philosophy and Social Sciences Planning Project (grant no. G2024-1-06) and Zhejiang Province Philosophy and Social Sciences Planning Project (grant no. 25NDJC136YB).

This research was supported by Research Project of Zhejiang Federation of Humanities and Social Sciences (grant no. 2025N161), Ningbo Philosophy and Social Sciences Planning Project (grant no. G2024-1-06) and Zhejiang Province Philosophy and Social Sciences Planning Project (grant no. 25NDJC136YB).

7. We note that you have indicated that there are restrictions to data sharing for this study. PLOS only allows data to be available upon request if there are legal or ethical restrictions on sharing data publicly. For more information on unacceptable data access restrictions, please see http://journals.plos.org/plosone/s/data-availability#loc-unacceptable-data-access-restrictions.

8. In the online submission form, you indicated that the data underlying the results presented in the study are available from authors.

Additional Editor Comments :

Opinions of the reviewers: one "major revision", one "major revision" and one "reject". I propose a major revision, but cannot predict the chances of final acceptance.

Reviewers' comments:

Reviewer's Responses to Questions

**Comments to the Author**

1. Is the manuscript technically sound, and do the data support the conclusions?

Reviewer #1: Yes

Reviewer #2: Yes

Reviewer #3: Partly

2. Has the statistical analysis been performed appropriately and rigorously? 

Reviewer #1: Yes

Reviewer #2: Yes

Reviewer #3: I Don't Know

3. Have the authors made all data underlying the findings in their manuscript fully available?

Reviewer #1: Yes

Reviewer #2: No

Reviewer #3: No

4. Is the manuscript presented in an intelligible fashion and written in standard English?

Reviewer #1: Yes

Reviewer #2: Yes

Reviewer #3: Yes

5. Review Comments to the Author

**Reviewer #1:**

This paper takes Ningbo Port, Shanghai Port, and Tangshan Port as the research objects, and selects the data from 2013-2022 to explore the inner mechanism of the impact of digital technology on the new-quality productivity of port logistics. The results and tests lead to some conclusions. The problem context is appealing, and the paper has some merits. However, overall, some issues still need to be improved.

1. The introduction lacks focus, innovation, and a concise structural overview.

2. A structural equation model (SEM) diagram is needed to visually represent variable relationships and hypotheses. Figure 1 does not align with the hypotheses in later sections; it should be redesigned for consistency with the theoretical framework.

3. “The regression models (4) and (5) purportedly validate H4”, but the hypothesis itself is not explicitly defined in the text. Ensure H4 is clearly stated and linked to these models.

4. Expand the variable descriptions to include selection criteria, and justification for their inclusion, a summary table is recommended. The explanation of Table 2 lacks coherence with its content. List specific foundational studies in section 3.2.3 “Referring to existing studies”.

5. Section 4.4.1: The exclusion of outlier years (2019–2020) to account for pandemic impacts is insufficient. Extend the period to 2020–2021, as port logistics were more severely disrupted during this time.

6. The dataset spans 2013–2022. Consider updating it to include the most recent data where feasible.

7. The current instrumental variable selection and methodology for endogeneity test lack empirical rigor. Reassess the approach using established econometric techniques to strengthen validity.

8. A comparative experiment involving the three ports is considered for addition to test whether differences among ports affect the conclusion.

9. Standardize citation formats throughout the manuscript and ensure consistency in reference list formatting.

10. Expand the discussion of Managerial insights, particularly regarding how port authorities and policymakers can leverage digital technologies to enhance new quality productivity. Provide actionable recommendations based on empirical findings.

**Reviewer #2:**

The manuscript offers valuable insights into the relationship between digital technology and productivity in port logistics. The authors have conducted a rigorous analysis with strong empirical support, but there are several areas where additional details and clarifications would improve the manuscript. After addressing these minor revisions, I would recommend this manuscript for publication.

1. The literature review primarily discusses the relationship between digital technology and productivity. However, it provides limited discussion on the specific measurement indicators of new-quality productivity. It is recommended to supplement relevant theoretical discussions to strengthen the conceptual foundation.

2. This manuscript employs multiple regression models, such as instrumental variable regression and fixed-effects regression. However, it lacks a detailed discussion on the economic rationale for choosing these models and their applicability. It is suggested to supplement relevant econometric methodological justifications.

3. This manuscript uses telephone penetration as an instrumental variable, but it lacks a detailed discussion on whether this variable satisfies the relevance and exogeneity assumptions.

4. The current model illustrations are relatively simplistic. It is recommended to include more intuitive causal pathway diagrams, such as how digital technology influences new-quality productivity through infrastructure and the digital economy. Adding more details would help readers better understand the research framework.

5. This manuscript selects three major ports—Ningbo, Shanghai, and Tangshan—as research objects, which are somewhat representative. However, further discussion is needed on whether the findings can be generalized to all ports in China or other logistics sectors. It is suggested to add explanations regarding the limitations of the dataset.

6. This manuscript cites a large number of Chinese references, but the formatting is inconsistent. For example, references [18], [19], and [27] are formatted differently. It is recommended to standardize the reference formatting for consistency.

**Reviewer #3:**

1.What is the innovative contribution of this paper?

2.The sample size is too small, and the data granularity is too coarse.

3.The metrics are not clearly defined. For example: What constitutes "New-Quality Productivity"? How is "Digital Technology" measured?

6. PLOS authors have the option to publish the peer review history of their article (what does this mean?). If published, this will include your full peer review and any attached files.

Reviewer #1: No

Reviewer #2: **Yes**

Reviewer #3: No

---

## [Author Response · Author response to Decision Letter 1]

2 Jun 2025

Dear Editor and Reviewers:

Thank you for your letter and for the reviewers’ comments concerning our manuscript entitled “Research on the Impact Mechanism of Digital Technology on the New-Quality Productivity of Port Logistics”(Manuscript ID: PONE-D-25-08375). Those comments are all valuable and very helpful for revising and improving our manuscript, as well as the important guiding significance to our research. We thank you very much for allowing us to revise our manuscript.

These feedbacks provide important guidelines for improving the quality of the article, and we have carefully sorted out and actively implemented the modifications, and now report the details as follows:

Reviewer1:

Q1: The introduction lacks focus, innovation, and a concise structural overview.

Response: We supplemented the limitations of existing research theories in the first paragraph of the introduction to clarify the research core of this paper: "First, the study mostly focuses on unidimensional improvements such as port transport efficiency, ignoring the systemic changes triggered by digital technologies; Second, it does not reveal the mediating role of digital infrastructure upgrades and the level of internalization of technology applications in the process of transformation of the new quality productivity”; the third paragraph is expanded on the basis of the original content to include a succinct overview of the structure, so that the logic of the introduction is more clear. (See lines 5 to 15 on page 3 and line 42 on page 3 to line 15 on page 4 for details)

Q2: A structural equation model (SEM) diagram is needed to visually represent variable relationships and hypotheses. Figure 1 does not align with the hypotheses in later sections; it should be redesigned for consistency with the theoretical framework.

Response: Thank you for valuable feedback. You pointed out the need for structural equation modeling diagrams to visualize the variable relationships and hypotheses, and pointed out the inconsistency between Figure 1 and the hypotheses, which is crucial for improving the presentation of the study. We have revised Figure 1 by adding two mediating variables in the new quality productivity of port logistics enabled by digital technology to make it consistent with the theoretical framework; meanwhile, we have added a new structural equation modeling diagram for the mechanism test analysis in Chapter 4.5, which presents in detail the composition of the evaluation indicators of digital technology and new quality productivity, as well as the influence coefficients among the explanatory variables, the explanatory variables and the mediating variables. (See lines 30 to 32 on page 8 and lines 8 to 9 on page 20 for details)

Q3: “The regression models (4) and (5) purportedly validate H4”, but the hypothesis itself is not explicitly defined in the text. Ensure H4 is clearly stated and linked to these models.

Response: Thank you for your suggestions. Regarding the issue of “H4 assumption is not clearly defined in the text”, we reorganize the correspondence between the model and the assumptions constructed in the text, and reset the model (1) to be linked to assumption H1, the model (2)(3) to be linked to assumption H2, and the model (4)(5) to be linked to assumption H3. This ensures the rigor of the article. (See lines 6 to 16 on page 14 for details)

Q4: Expand the variable descriptions to include selection criteria and justification for their inclusion; a summary table is recommended. The explanation of Table 2 lacks coherence with its content. List specific foundational studies in section 3.2.3 “Referring to existing studies”.

Response: Based on your suggestions, we have added variable descriptions and rationale for the selection of primary and secondary indicators before sections 3.2.1 and 3.2.2:” The first-level indicator adopts intangible labor data to reflect the intensity of the port's investment in intangible assets such as patents, software and brands, which is the core characterization of innovation-driven productivity; fixed labor data directly reflects the comprehensive processing capacity of the port logistics system, which also affects the process of automation and intelligent transformation; high-quality talents are the foundation of the human capital to promote the digital transformation and technological innovation; the financial risk indicator reflects the enterprise's investment The financial risk indicator reflects the enterprise's investment capacity, which affects the introduction of new technologies and facility upgrading; the future development indicator reflects the agility of the port logistics system and its ability to optimize the operation of the existing assets, which is highly compatible with the cost reduction and efficiency enhancement goals required by the new quality productivity.”; We added columns for indicator selection criteria in Tables 1 and 2; And we added references to existing studies in section 3.2.3 to make the variable-related content more detailed. (See lines 30 to 32 on page 8 and lines 8-9 on page 20 for details)

Q5: Section 4.4.1: The exclusion of outlier years (2019–2020) to account for pandemic impacts is insufficient. Extend the period to 2020–2021, as port logistics were more severely disrupted during this time.

Response: Thank you for your valuable proposal. We fully agree with your request to extend the period of excluded anomalous years to 2020 - 2021. The excluded anomalous years have been adjusted to 2020 - 2021 in Section 4.4.1 to more reasonably account for the shocks received by port logistics. (See lines 3 to 7 on page 17 for details)

Q6: The dataset spans 2013–2022. Consider updating it to include the most recent data where feasible.

Response: Thank you for your careful review and valuable suggestions on our manuscript. We re-collected data for the year 2023 and updated the full-text data to ensure that the study data is current. (See lines 19 to 23 on page 14 and line 1 on page 15 for details)

Q7: The current instrumental variable selection and methodology for the endogeneity test lack empirical rigor. Reassess the approach using established econometric techniques to strengthen validity.

Response: We are very grateful for your valuable proposal. In the endogeneity test, we re-selected the breadth of digital technology application as an instrumental variable, referring to Yang Peng's approach. Yang Peng used text analysis to search, match, and count the word frequency of keywords related to “digital technology” in the annual reports of listed companies according to the breadth of digital technology application, and expanded the sample in the year of study. We used established econometric techniques to re-evaluate the methodology and enhance the empirical rigor of the study. (See lines 4 to 24 on page 16 and line 1 on page 17 for details)

Q8: A comparative experiment involving the three ports is considered for addition to test whether differences among ports affect the conclusion.

Response: Thank you for your careful review and valuable suggestions. Your suggestion to add a comparative experiment involving three ports is of great academic value and practical guidance for exploring in depth how differences between ports affect research findings. Through the comparative experiment, the impact of the differences in geographic location, scale of operation, and policy environment on the relationship between digital technology and new quality productivity can be more clearly identified, thus making the research conclusions more generalizable and in-depth. However, due to the limitations of research time and data acquisition, we regret that we are unable to conduct this experiment for the time being. We will incorporate this into our research program as soon as conditions permit.

Q9: Standardize citation formats throughout the manuscript and ensure consistency in reference list formatting.

Response: We have overhauled the formatting of the bibliographic citations to ensure that the citation formatting and the formatting of the reference lists are consistent throughout the manuscript.” Burinskiene A, Daskevic D. The investigation on the application of digital technologies for logistics business competitiveness[J]. Tehnički glasnik, 2024, 18(4): 626-637.”. (See lines on page 23 to page 26 for details)

Q10: Expand the discussion of Managerial insights, particularly regarding how port authorities and policymakers can leverage digital technologies to enhance new quality productivity. Provide actionable recommendations based on empirical findings.

Response: Thank you for your valuable suggestions. In conjunction with the empirical results, we further expand the discussion of management insights by providing actionable recommendations for port authorities and policymakers at three levels. (See lines 15 to 44 on page 21 and lines 1 to 5 on page 22 for details)

Thank you again for your positive and constructive comments and suggestions on our manuscript.

Reviewer2:

Q1: The literature review primarily discusses the relationship between digital technology and productivity. However, it provides limited discussion on the specific measurement indicators of new-quality productivity. It is recommended to supplement relevant theoretical discussions to strengthen the conceptual foundation.

Response: Thank you for your valuable proposal. Your suggestion that the literature review has limited discussion of specific measures of NQP is critical. We have added a theoretical discussion of digital technologies and specific measures of NQP in Section 3 of the article to strengthen the conceptual foundation. The level 1 indicator adopts intangible labor data to reflect the intensity of the port's investment in intangible assets such as patents, software, brands, etc., which is the core characterization of innovation-driven productivity; the fixed labor data directly reflects the comprehensive processing capacity of the port logistics system, which also affects the process of automation and intelligent transformation; high-quality talents are the human capital basis for promoting digital transformation and technological innovation; the financial risk indicator reflects the enterprise's investment The financial risk indicator reflects the enterprise's investment capacity, which affects the introduction of new technologies and facility upgrading; the future development indicator reflects the agility of the port logistics system and its ability to optimize the operation of the existing assets, which is highly compatible with the cost reduction and efficiency enhancement goals required by the new quality productivity. (See lines 30 to 32 on page 8 and lines 8 to 9 on page 20 for details)

Q2: This manuscript employs multiple regression models, such as instrumental variable regression and fixed-effects regression. However, it lacks a detailed discussion on the economic rationale for choosing these models and their applicability. Supplementing relevant econometric methodological justifications is suggested.

Response: In response to your comments, we have included a rationale for selecting the baseline regression analysis model and a discussion of model fit in Section 3.3, which details the econometric methodological rationale. This study adopts a baseline regression analysis model, which can effectively control individual heterogeneity, such as port location advantage and policy additions, to avoid estimation errors caused by omitted variables. There may be a bidirectional causal relationship between digital technology and new quality productivity of port logistics, and the use of a fixed-effects model can eliminate individual effects that do not change over time and alleviate some endogenous disturbances through demeaning treatment. (See lines 13 to 14 on page 13 and lines 1 to 4 on page 14 for details)

Q3: This manuscript uses telephone penetration as an instrumental variable, but it lacks a detailed discussion on whether this variable satisfies the relevance and exogeneity assumptions.

Response: In the endogeneity test, we re-selected the breadth of digital technology application as an instrumental variable, referring to Yang Peng's approach. Yang Peng used text analysis to search, match, and count the word frequency of keywords related to “digital technology” in the annual reports of listed companies according to the breadth of digital technology application, and expanded the sample in the year of study. We used established econometric techniques to re-evaluate the methodology and enhance the empirical rigor of the study. (See lines 4 to 24 on page 16 for details)

Q4: The current model illustrations are relatively simplistic. It is recommended to include more intuitive causal pathway diagrams, such as how digital technology influences new-quality productivity through infrastructure and the digital economy. Adding more details would help readers better understand the research framework.

Response: Thank you for your valuable suggestion. We have revised Figure 1 by adding two mediating variables in the new quality productivity of port logistics enabled by digital technology to make it consistent with the theoretical framework; meanwhile, we have added a new structural equation modeling diagram for the mechanism test analysis in Chapter 4.5, which presents in detail the composition of the evaluation indicators of digital technology and new quality productivity, as well as the influence coefficients among the explanatory variables, the explanatory variables and the mediating variables. (See lines 30 to 32 on page 8 and lines 8 to 9 on page 20 for details)

Q5: This manuscript selects three major ports—Ningbo, Shanghai, and Tangshan—as research objects, which are somewhat representative. However, further discussion is needed on whether the findings can be generalized to all ports in China or other logistics sectors. It is suggested to add explanations regarding the limitations of the dataset.

Response: Your discussion of the generalizability of the study findings and the limitations of the dataset is extremely valuable. Due to research resources and time constraints, a more in-depth discussion of dataset limitations is not possible at this time. In subsequent studies, we will further expand the scope of the study and refine our analysis of this issue.

Q6: This manuscript cites a large number of Chinese references, but the formatting is inconsistent. For example, references [18], [19], and [27] are formatted differently. It is recommended to standardize the reference formatting for consistency.

Response: We have overhauled the citation formatting to ensure that the citation formatting and the formatting of the reference list are consistent throughout the manuscript. [18] Fan S. Influencing factors and countermeasures on intelligent transformation and upgrading of logistics firms: A case study in China[J]. Plos one, 2024, 19(4): e0297663. [19] Burinskiene A, Daskevic D. The investigation on the application of digital technologies for logistics business competitiveness[J]. Tehnički glasnik, 2024, 18(4): 626-637. [27] Zhao Huida, Liu Jiaguo, Hu Xiyuan. Servitization with blockchain in the maritime supply chain[J]. Ocean & Coastal Management, 2022, 225: 106195.( See lines on page 23 to page 26 for details)

Thank you again for your positive and constructive comments and suggestions on our manuscript.

Reviewer3:

Q1: What is the innovative contribution of this paper?

Response: Thank you very much for your request to clarify the innovative contribution, which helps readers better understand the research value. This paper breaks through the traditional technological instrumentalism perspective, constructs the theoretical framework of "digital technology penetration - new digital infrastructure empowerment - technology application internalization - productivity transformation", and systematically reveals the intrinsic mechanism of digital technology influencing the new quality of port logistics. This paper breaks through the traditional technological instrumental perspective to construct the theoretical framework of "digital technology penetration - new digital infrastructure empowerment - technology applicatio

---

## [Decision Letter · Decision Letter 1]

28 Jul 2025

PONE-D-25-08375R1Research on the Impact Mechanism of Digital Technology on the New-Quality Productivity of Port LogisticsPLOS ONE

Dear Dr. Yuan,

Thank you for submitting your manuscript to PLOS ONE. After careful consideration, we feel that it has merit but does not fully meet PLOS ONE’s publication criteria as it currently stands. Therefore, we invite you to submit a revised version of the manuscript that addresses the points raised during the review process.

We look forward to receiving your revised manuscript.

Kind regards,

Xu Xin

Academic Editor

PLOS ONE

Journal Requirements:

Reviewers' comments:

Reviewer's Responses to Questions

**Comments to the Author**

1. If the authors have adequately addressed your comments raised in a previous round of review and you feel that this manuscript is now acceptable for publication, you may indicate that here to bypass the “Comments to the Author” section, enter your conflict of interest statement in the “Confidential to Editor” section, and submit your "Accept" recommendation.

Reviewer #1: (No Response)

Reviewer #2: All comments have been addressed

2. Is the manuscript technically sound, and do the data support the conclusions?

Reviewer #1: (No Response)

Reviewer #2: Yes

3. Has the statistical analysis been performed appropriately and rigorously? 

Reviewer #1: (No Response)

Reviewer #2: Yes

4. Have the authors made all data underlying the findings in their manuscript fully available?

Reviewer #1: (No Response)

Reviewer #2: Yes

5. Is the manuscript presented in an intelligible fashion and written in standard English?

Reviewer #1: (No Response)

Reviewer #2: Yes

6. Review Comments to the Author

**Reviewer #1:**

This paper takes Ningbo Port, Shanghai Port, and Tangshan Port as the research objects, and selects the data from 2013-2022 to explore the inner mechanism of the impact of digital technology on the new-quality productivity of port logistics. The results and tests lead to some conclusions. The problem context is appealing, and the paper has some merits. However, overall, some issues still need to be improved.

1. The innovation need to be improved.

2. Standardize citation formats throughout the manuscript and ensure consistency in reference list formatting.

3. Please adjust the table formats to ensure that the tables are clearly presented.

**Reviewer #2:**

I have no further comments. The authors have made the revisions and responses according to my comments, and the paper can be accepted.

7. PLOS authors have the option to publish the peer review history of their article (what does this mean?). If published, this will include your full peer review and any attached files.

Reviewer #1: No

Reviewer #2: **Yes: **Chao Mi

---

## [Author Response · Author response to Decision Letter 2]

2 Aug 2025

Dear Editor and Reviewers:

Thank you very much for your valuable feedback on our manuscript entitled “Research on the Impact Mechanism of Digital Technology on the New-Quality Productivity of Port Logistics”. (Manuscript ID: PONE-D-25-08375). These feedbacks provide important guidelines for improving the quality of the article, and we have carefully sorted out and actively implemented the modifications, and now report the details as follows:

Reviewer1:

Q1: The innovation need to be improved.

Response: To enhance the innovation of the article, we have made revisions in several aspects: In terms of research topic selection, the introduction emphasizes the lack of mechanistic exploration in existing studies on the "systematic driving effect of digital technology on new-quality productivity in port logistics". (See lines 4 to 6 on page 4 for details) We innovatively take "digital technology" and "new-quality productivity in port logistics" as core interrelated objects, focusing on their intrinsic action logic. (See lines 3 to 5 on page 17 for details) In terms of research methods, we supplemented the description of the explained variable by clarifying that the evaluation index of new-quality productivity in port logistics is constructed from multiple dimensions, which improves the scientificity of variable measurement. Additionally, a two-way causality test was added in the endogeneity test to further verify the robustness of the conclusions. (See line 1 on page 25 to line 1 on page 26 for details)

Q2: Standardize citation formats throughout the manuscript and ensure consistency in reference list formatting.

Response: As requested, we have thoroughly standardized the citation formats throughout the entire manuscript and ensured strict consistency in the reference list. Specifically, the revisions include: in the manuscript, the formatting of authors' names has been kept consistent. In the references section, the formatting of authors' names has been standardized, and publication information such as journal volume, issue, and page ranges has been uniformly presented, with punctuation and spacing ensured to be standardized and consistent. (See line 1 on page 36 to line 10 on page 40 for details)

Q3: Please adjust the table formats to ensure that the tables are clearly presented.

Response: Thank you for your feedback on the table formatting. We have carefully adjusted the formatting of all tables in the manuscript to enhance clarity. Specifically, we have standardized the alignment of columns, optimized border lines, and ensured consistent spacing between cells and text, thereby improving readability. These revisions aim to make the tables easier to understand while being consistent with the overall layout of the manuscript. (See table 1 on page 16 and 17, table 2 on page 18 and 19, table 3 on page 21, table 4 on page 22 and 23, table 5 on page 24 and 25, table 6 on page 25 and 26 for details)

We tried our best to improve the manuscript and made some other changes in the manuscript. Here we did not list the changes but marked the changed parts in red in the revised version.

Once again, we sincerely thank the editor and reviewers for their valuable comments, which are of great significance in improving the quality of this paper. If you have any other questions or suggestions about the revision, please feel free to point them out, and we will continue to improve it.

---

## [Decision Letter · Decision Letter 2]

9 Sep 2025

Research on the Impact Mechanism of Digital Technology on the New-Quality Productivity of Port Logistics

PONE-D-25-08375R2

Dear Dr. Yuan,

We’re pleased to inform you that your manuscript has been judged scientifically suitable for publication and will be formally accepted for publication once it meets all outstanding technical requirements.

Kind regards,

Xu Xin

Academic Editor

PLOS ONE

Reviewers' comments:

Reviewer's Responses to Questions

**Comments to the Author**

1. If the authors have adequately addressed your comments raised in a previous round of review and you feel that this manuscript is now acceptable for publication, you may indicate that here to bypass the “Comments to the Author” section, enter your conflict of interest statement in the “Confidential to Editor” section, and submit your "Accept" recommendation.

Reviewer #1: All comments have been addressed

Reviewer #2: All comments have been addressed

2. Is the manuscript technically sound, and do the data support the conclusions?

Reviewer #1: Yes

Reviewer #2: Yes

3. Has the statistical analysis been performed appropriately and rigorously? 

Reviewer #1: Yes

Reviewer #2: Yes

4. Have the authors made all data underlying the findings in their manuscript fully available?

Reviewer #1: Yes

Reviewer #2: Yes

5. Is the manuscript presented in an intelligible fashion and written in standard English?

Reviewer #1: Yes

Reviewer #2: Yes

6. Review Comments to the Author

Reviewer #1: The author has addressed all the comments and completed the revisions. I recommend that the manuscript be accepted.

Reviewer #2: The author has made revisions according to all reviewers' requests and I believe the manuscript can be accepted.

7. PLOS authors have the option to publish the peer review history of their article (what does this mean?). If published, this will include your full peer review and any attached files.

Reviewer #1: No

Reviewer #2: **Yes**

---

## [Editor Report · Acceptance letter]

PONE-D-25-08375R2

PLOS ONE

Dear Dr. Yuan,

I'm pleased to inform you that your manuscript has been deemed suitable for publication in PLOS ONE. Congratulations! Your manuscript is now being handed over to our production team.

Kind regards,

on behalf of

Dr. Xu Xin

Academic Editor

PLOS ONE